# Nursing Home and Vaccination Consent: The Italian Perspective

**DOI:** 10.3390/vaccines9050429

**Published:** 2021-04-24

**Authors:** Nunzia Cannovo, Roberto Scendoni, Marzia Maria Fede, Federico Siotto, Piergiorgio Fedeli, Mariano Cingolani

**Affiliations:** 1Federico II University Ethics Committee, 80131 Naples, Italy; nunzia.cannovo@gmail.com; 2Law Department, Legal Medicine Section, Macerata University, 62100 Macerata, Italy; roberto.scendoni@gmail.com (R.S.); marziamaria.fede@gmail.com (M.M.F.); 3School of Law, Labour Law, Camerino University, 62032 Camerino, Italy; federico.siotto@unicam.it; 4School of Law, Legal Medicine, Camerino University, 62032 Camerino, Italy; piergiorgio.fedeli@unicam.it

**Keywords:** Covid-19 vaccination, elderly vaccination, informed consent to vaccination, capacity to consent

## Abstract

Since the beginning of the Covid-19 pandemic, many countries have begun vaccination campaigns, with different methods and timelines, with the goal of vaccinating over 75% of the population and thus achieving herd immunity. Initially it was necessary to identity the categories of citizens who should be the first to receive the vaccines, on the basis of scientific evidence. On the basis of this information, elderly residents in nursing homes and the staff who care for them should be the highest priority subjects for vaccination. In this context, obtaining informed consent to Covid-19 vaccination presents a considerable challenge, as the advanced age and frequent comorbidities of a significant number of the residents may mean that they are incapable of expressing consent themselves. The legislation of various Western nations substantially agrees on the general principle that those capable of judgement must be asked for their consent for healthcare services, and that even those with psychological weaknesses that limit their full ability to decide must be involved in these decision-making processes. The article can help systematize the processes to be implemented to protect the health of individuals as members of a close and fragile community.

## 1. Introduction

Since the beginning of the Covid-19 pandemic, worldwide efforts to contain its effects have focused on research [1] to develop vaccinations, often through synergy between the public and private sectors. This objective has been achieved in record time [2,3] while respecting all the required phases for authorization to put the vaccinations on the market, under the aegis of the various drug agencies (EMA European Medicine Agency, FDA Food and Drugs Administration, etc.) [4,5,6,7,8].

Many countries have begun vaccination campaigns, with different methods and timelines, with the goal of vaccinating over 75% of the population and thus achieving herd immunity [9].

In this context, most Western nations have prioritized the order in which various categories of citizens should be vaccinated [10].

Initially, the number of vaccine doses was insufficient to vaccinate the entire population, and thus it was necessary to identity the categories of citizens who should be the first to receive them 10, on the basis of scientific evidence. Health was identified as the first right to be safeguarded and thus the first criterion for identifying the highest priority categories.

Very strong scientific evidence indicated that the risk of death from Covid-19 increased exponentially with the increase in age [11,12,13].

In addition, epidemiological studies pointed to the fact that residents of nursing homes were the category worst hit by Covid-19 infection, first, because these community settings inherently present higher risk of exposure to the infection, and second, because residents already have comorbidities and thus are already at higher clinical risk for grave illness and death [14,15,16].

On the basis of this information, elderly residents in nursing homes and the staff who care for them should be the highest priority subjects for vaccination. In fact, staff in residential nursing homes have been shown to have higher infection rates than workers who provide assistance in private homes [14,15,16]. Thus, the twofold goal of vaccinating residential nursing home staff is to avoid having asymptomatic but contagious infected staff members spread the disease among residents, and also to prevent negative repercussions on healthcare in nursing homes caused by staff absences due to illness.

## 2. The State of the Art on Europe

The legislation on vaccination campaigns developed by many nations is substantially uniform in identifying the residents and workers of nursing homes as very high priority subjects to be vaccinated, on the basis of scientific evidence. The following is the position of some European states, which together are representative of all the possible situations that exist in the remaining member states.

The United Kingdom’s 25 September 2020 independent report “JVCI Advice on Priority Groups for Covid-19 Vaccination”, updated numerous times [17], most recently on 5 January 2021 [18] identified the following priorities for the administration of the vaccine in England, Wales, and Northern Ireland [19]
–those aged 80 and up;–those who live or work in nursing homes;–healthcare and social workers at high risk;–and, for Scotland, it dictated vaccination for “residents in a nursing home for the elderly and their carers”.

In Italy, the Strategic Plan for “Anti-SARS-CoV-2/COVID-19 Vaccination [20]” established priority vaccination for “Residents and staff of residential nursing homes for the elderly” with the logistics to be organized by the Health Ministry [21].

In Spain, the National Healthcare System initially formed a working group that wrote its “*Estrategia de vacunación frente a COVID19 en España* [22]”, establishing the priority groups for vaccination, at the top of which was “*Residents and healthcare and social workers in nursing homes for the elderly and/or disabled. If necessary, priority goes to vaccinations for the most vulnerable residences (the greatest number of prisoners, poorest ability to adopt prevention and control measures and/or residences that have not had Covid-19 cases*)”.

In France, in the initial phase of the Covid19 pandemic, the National Authority on Healthcare [23] prioritized two populations for vaccination on the basis of the parame ters of vulnerability (age and/or comorbidities) and risk of greater exposure to the virus:(1)Residents of nursing home structures and residents in long-term care facilities (EHPAD, USLD, etc.);(2)Professionals who work in institutions for the elderly (mainly EHPAD and USLD), also at greater risk of severe forms of infectious disease or death (those over 65 and/or with comorbidities).

The same priority was substantially established in other nations of the European Union and in most parts of the United States [24].

Thus, there is a unanimous opinion of residents and staff in nursing homes for the elderly as those who should be the first to be vaccinated, given that the residents belong to the oldest group and frequently have comorbidities that increase their vulnerability to Covid 19 infection [25,26].

## 3. The Problem of Informed Consent

In this context, obtaining informed consent to Covid-19 vaccination presents a considerable challenge, as the advanced age of nursing home residents and their frequent comorbidities may result in the inability to express their own consent.

In fact, precisely this difficulty of acquiring consent is complicating the urgent work to vaccinate these subjects throughout the world. Many have lamented that even though governments knew vaccines would be available in December 2020 and that nursing home residents and staff would be among the first scheduled to receive vaccinations, they failed prepare in good time the mechanisms for proceeding correctly for this category [27].

This has led to disparities in interpretation of the general directives and consequently disparities in implementation; in this context, it was not possible to “wait and see,” given the need to vaccinate with all speed in order to safeguard the health of the individuals and other members of the close community of nursing homes.

The legislation of various Western nations, drawing upon the Oviedo Convention [28], substantially agrees on the general principle that those who have the capacity to consent to vaccination must be asked for their consent for healthcare services, and that even those with limited or compromised capacity that limit their full ability to decide must be involved in these decision-making processes [29,30,31,32,33,34].

Many challenges can arise in seeking to obtain informed consent from nursing home residents. Subjects who are capable of judgement, or their legal representatives, may refuse consent for vaccination. Similarly, assessing whether persons incapable of forming judgments are nevertheless capable of giving specific consent to vaccination may prove to be a complex challenge. In addition, some residents who are incapable of providing consent may not have previously chosen or been assigned someone as their legal guardian.

In short, the following situations may arise:Mentally competent residents who provide consent;Mentally competent residents who refuse consent;Mentally incompetent residents whose Court of Protection-appointed deputy (in Italy there are two kinds of deputies, a “tutore” or an “amministratore di sostegno,” the latter if specifically designated for health issues in the decree of appointment of deeds) provides consent;Mentally incompetent residents whose Court of Protection-appointed deputy (in Italy there are two kinds of deputies, a “tutore” or an “amministratore di sostegno,” the latter if specifically designated for health issues in the decree of appointment of deeds) refuses consent;Subjects whose capacity to provide consent must be ascertained;Subjects who are obviously mentally incompetent, for whom a legal order has not yet been made to remove their legal authority to make healthcare decisions for themselves;

In situations (1) and (3), there are no operational problems.

Case (2) presents a dilemma, when a nursing home resident with capacity refuses the vaccination.

This is a concrete possibility in Western nations, as reported in some studies conducted in the United States [35,36], but not so in China, where a high number of people accepted vaccination [37]. As the Italian National Bioethics Commission has stressed, it is essential to work to provide the public with the necessary information before launching a vaccination campaign [38].

Given that Covid-19 vaccination is not obligatory in the nations examined, one solution could be to organize “Anti-Vax” wards for residents who are not self-sufficient and have no other place (family) to go. Should the nursing home be unable to provide such a ward, the “anti-vax” resident ought to be transferred to another structure, for the benefit of the vaccinated residents who then might be able to have visitors once again.

Clearly, the residence where to move the “anti-vax” resident must ensure the protection of all rights of the person, in accordance with the provisions of Italian law and the “Convention on the Rights of Persons with Disabilities (CRPD)” of the United Nations. The opportunity to welcome friends and family to the nursing home once again would be precluded if unvaccinated residents were to remain there.

For situation (4), the same arrangements could be made as for situation (2), even though in Italy recent legislation has opened other scenarios. The Legislative Decree of 5 January 2021, Section 4, article 5 [39], states that if the legal representative of the subject refuses consent for vaccination, the healthcare authorities could appeal to the Court of Protection for permission to do the vaccination. This law was written to promote vaccination of those residing in assisted healthcare structures who are deemed incompetent to make decisions for themselves.

Residents who are weak-minded or have a fleeting memory due to initial cognitive impairment present a decidedly more complex situation, as do those who clearly are incapable of making their own decisions but for whom no legal provision has been made regarding the itinerary to follow to obtain consent for healthcare services in their own best interests.

It is well known that there can be different levels of compromised ability to consent to or refuse healthcare treatments. For many nations, the point of reference in these cases is the Oviedo Convention, Chapter II, which deals with consent. The subject should always be involved in the decision-making process, even if limited by mental deficits that have a negative impact on the process.

Many contributions have been written on this subject [40,41,42,43,44,45,46,47]. With specific reference to vaccination, of particular interest is the Alzheimer’s Society (United Against Dementia) article on “*Consent to coronavirus (COVID-19) vaccination*” updated 22 December 2020 [48] published from the same United Kingdom care and research charity. Also important in this regard is the 24 December 2020 guide published by the UK Department of Health & Social Care, “The Mental Capacity Act (2005) (MCA) and deprivation of liberty safeguards (DoLS) during the coronavirus (COVID-19) pandemic: additional guidance”, which states that all possible measures should be taken to support the person in deciding whether or not to be vaccinated. It refers readers to the operative indications provided in the Mental Capacity Act, which require that the person who must make the final decision should consider all the relevant circumstances, including the desires, convictions and values of the person and the opinions of his or her family, seeking to understand what the person would have chosen had she or he been capable. Though the decision affects higher interests, it should nonetheless be made on an individual basis, and one should not automatically assume that the best interests of the person are also in the best interests of others.

The goal is to ascertain whether the person in possession of his or her full faculties would have chosen to be vaccinated against Covid-19.

Clearly, it might be hypothesized that most people who consciously decided to move into a nursing home would have accepted vaccination for the protection of their own health and that of those around them, but this acceptance is not a given. In this case, the involvement of family members is important.

In the Italian legal system, the competence indicates the ability of the subject to perform legal acts through which he/she acquires rights and assumes duties and this presupposes the full capacity to make decisions. However, the global decision-making impairment does not always correspond to the concept of decisional capacity, since a subject may not be able to take a wide range of decisions in daily-life activities (financial matters, nursing home admittance, contracts, etc.), but retain the capacity to make autonomous health-care decisions.

As quoted in literature [49], a patient has medical decision-making capacity if he/she can demonstrate understanding of the situation, the consequences of his/her decision, and reasoning in his/her thought process, and if he/she can communicate his/her wishes. The absence of one of these abilities does not mean that a patient lacks the overall ability to make a decision.

A patient can have the capacity to make small, straightforward decisions such as consenting to take a vaccine, but may lack “the capacity to consent to a high-risk abdominal surgery [50]”.

It has long been stated in the literature that decision-making capacity, medical or otherwise, is always specific to the task requiring the decision [51,52,53,54]. Literature has proposed a level of required capacity on the decision to be made (higher for critical ones, lower for low-risk decisions) [55,56].

Italy’s Legislative Decree of 5 January 2021, article 5 [39] provides a detailed procedure for obtaining vaccination consent from mentally incompetent subjects. The legislators without a doubt favored the decision-making process that promotes a positive conclusion in terms of consent to vaccination.

In particular, when it must be ascertained whether the subject is capable of self-determination in healthcare decisions (consent to vaccination), the physician must certainly explore the residual mental capacities, most importantly the comprehension of the information received, the capacity to evaluate the alternative behavior and the related consequences, and the capacity to express a choice [57]. This diagnostic process need not necessarily include psychometric tests, but must seek to involve family members when possible. While it is not within the purview of this article to describe in detail the difficulty to implement in practice, it is important to stress the importance of involving the subject and his or her family, even if the final decision will be made by the person appointed by the Court of Protection. It is advisable to keep written records of the process of obtaining consent, on the basis of the reference legislation of various nations. 

Traditionally, international legislation in the presence of a cognitively impaired individual provides for surrogate decision makers. In literature [57], for some years now, it has been considered an alternative to plenary guardianship and substitute decision making in at least some cases. According to Davidson et al., Supported Decision Making (SDM) refers to the process of supporting people, whose decision-making ability may be impaired, to make decisions and so promote autonomy and prevent the need for substitute decision making [58]. In SDM, the patient with cognitive disabilities receives assistance from family, friends, or other trusted persons to enhance their decision-making capacity and skill so that they may retain autonomy during the decision-making process [59,60], thus implementing the principles set out in art. 12 of the Convention on the Rights of Persons with Disabilities (CRPD) [61,62].

The majority of international normative systems are still structured according to the masculine concepts that the individual alone should exercise his autonomy without the assistance of anyone, and if he is unable to do so, there should be a third person to express himself. Instead, the social model envisioned by the application of the CRPD would require that the legislation promote by all means that an individual with cognitive disabilities be able to participate in decision-making, either independently or with the assistance of others. Arstein-Kerslake et al. [63] recently noted that “more evidence needs to be gathered on how to meaningfully and accurately discover an individual’s will and preference and how that process can become part of service provision and other more formal structures”.

Thus, the following processes could be envisaged:(1)Mentally competent residents who provide consent vaccination;(2)Mentally competent residents who refuse consent: transfer to an “anti-vax” section of the same nursing home, or to a dedicated structure for subjects who refuse vaccination, where their contact with people outside the structure will be restricted;(3)Mentally incompetent residents whose Court of Protection-appointed guardian provides consent: vaccination.(4)Mentally incompetent residents whose Court of Protection-appointed guardian refuses consent: those who have the right can decide whether to appeal to the Court of Protection for its intervention;(5)Subjects whose capacity to provide consent must be ascertained: evaluate the residual capacities of the subject in terms of decision-making, seeking to reach a favorable, shared decision involving family members;(6)Subjects who are obviously mentally incapable, for whom a legal order has not yet been made to remove their legal authority to make healthcare decisions for themselves: following the legislation of the individual nations, taking into consideration any comorbidities, proceed with vaccination;

In this context, an extremely recent Italian Decree of 1 April 2021 is worth mentioning. Art. 5 amends art. 1: quinquies of the decree law of 18 December 2021 converted by Law No. 6 of 29 January 2021. Specifically, paragraph 2-bis is added, which states: “When the person in a state of natural incapacity is not hospitalized at health care facilities or similar facilities, however named, the functions of the support administrator, for the sole purpose of providing the consent referred to in paragraph 1, are carried out by the health director of the ASL of assistance or his delegate”. In essence, it has filled the legislative gap that omitted the people in a state of natural disability not hospitalized in healthcare facilities, for which, thanks to the new provision, will be followed under the same procedure as the people hospitalized in these facilities, as established by paragraph 2, for the sole purpose of consent to vaccination against COVID-19 [64].

## 4. Vaccination of Nursing Home Workers

In Italy, there is no law for making vaccine against Covid-19 mandatory and, as a consequence, there is no legal obligation for workers. Despite the lack of a general rule, we can call into question whether there is an obligation by reading the special provisions on the employment relationship. The interpretations are varied and differentiated, however they are concentrated in two main orientations: on the one hand, there is room for making vaccinations compulsory in the company by looking at articles 2087 Civil Code as well as 20 and 279 of Legislative Decree No. 81/2008, because the employer has a legal obligation to ensure health and safety at work; on the other hand, according to Article 32, paragraph 2, of the Italian Constitution, only a law can force a person to undergo health treatment: this perspective is enforced by articles 5 and 8, of Law No. 300/1970 (the so-called Charter of Workers’ rights) that clashes with the first option.

## 5. The Role of Compulsory Vaccination

A central issue is whether or not vaccination is compulsory for healthcare staff in assisted-living facilities. There is a very high probability (rebuttable presumption) for healthcare workers that they may come into contact with coronavirus, aside from their hospital ward, as well as for non-healthcare staff working in the hospitals for technical support, cleaning or aides, etc. The assisted healthcare residence staff are exposed to a high risk of occupational illness. During the pandemic phase, this means that the specific work carried out by a nurse is in itself a factor that increases the risk of Covid-19 infection. For this reason, the role of compulsory vaccination cannot be underestimated, except in cases of refusal justified by medical reasons or dangerous situations endangering worker’s health and safety (Art. 44 of the Legislative Decree no. 81/2008). The lack of vaccination, however, would make the worker’s service not useful for employers, especially in health and social care contexts where employees are in close contact with patients. This is a circumstance of supervening impossibility of performing conditions (or partial impossibility *ratione tempuras*); as a consequence, by applying the articles 1256, paragraph 2, and 1464 of the Civil Code we should balance the employer’s interest to layoff against the employee’s interest in maintaining employment. Following the refusal to be vaccinated by the employee the employer may refuse the worker performance and, consequently, the suspension of the employment relationship without pay. Moreover, the lack of a specific provision to ensure a general obligation to vaccinate is not a sufficient argument to exclude any liability on the employee side in the context of health services such as assisted healthcare residence. With regard to this specific context, the most cautious approach is to recommend vaccinations or, more incisively, to impose an additional burden to the worker in this particular sector, this means introducing another obligation in order to achieve a specific result: maintaining the employment relationship and, at the same time, reducing other risks out of the employment relationship related to social contact with other workers or other third parties (patients and their families).

To support the latter approach, we can refer to article 1 of Law No. 24/2017, which indirectly ensures every patient the right to safe care. The same provision imposes an obligation of providing healthcare treatments safely and avoiding any foreseeable related risk to public and private health facilities as well as to all practitioners of health professions. Article 1, paragraph 2, Law No. 24/2017 states that “the safety of care is also achieved through the set of all activities aimed at the prevention and management of risk” and that (paragraph 3) “the risk prevention activities implemented by public private health care facilities, is required to contribute all employees, including professionals affiliated”.

## 6. Recent Case Law and New Legislation

The vaccine against Covid-19 gives us complex problems with regard to the employment relationship, especially looking at some workplaces such as the assisted-living facilities. The dilemma between obligations based on individual consent and obligations based on reciprocity arises again. Although the tendency is to believe that acquiring consent is enough to avoid legal complaints, it is difficult to make ethical sense of community life if anyone does not recognize the power of solidarity (collective interest).

Recently, the Court of Belluno has ruled by order of 19 March 2021 on the appeal brought by some nursing home workers who have refused the COVID-19 vaccine despite being at their disposal and were put on paid leave by the employer. The judge considered legitimate leave and rejected the appeal arguing that now the vaccine is able to protect the physical integrity of individuals to whom it is administered, preventing the spread of disease and given the high risk in which residents of such facilities incur, the employer has an obligation to protect employees under Article 2087 of the Civil Code and therefore it is legitimate to change a job or remove the employee who will return once vaccinated. However, it is questionable what decision to take once the vacation period is over. Precisely on this issue, the decree of 1 April 2021, provided in art. 4, that all workers in the health sector are obliged to undergo anti-Covid vaccination, which is an essential requirement for the exercise of the profession and for the performance of work services rendered by the obliged parties. The refusal of the worker determines the suspension, without pay, from the right to perform services or tasks at risk of spreading Covid-19 infection until the fulfillment of the vaccination obligation or, failing that, until the completion of the national vaccination plan and in any case not later than 31 December 2021.

## 7. Conclusions

In conclusion, the beginning of the Covid-19 pandemic found the world community unprepared, and efforts to deal with it have been varied and uncoordinated, both by individual nations and by international organizations such as the United Nations, World Health Assembly, European Union, United Kingdom, United States. After the WHO declaration of the pandemic nature of the Covid-19 infection in January 2020, each state has applied very different measures to contain the contagion.

Unfortunately, even when it was possible to foresee critical situations and to organize appropriate systems in good time, on all levels there was lack of diligence that only deepened crises and produced risks for the health of individuals and the community. In Italy, which was the European country first affected by the infection, it was the one in which the incidence of mortality was highest. The lack of attention to strict containment measures relating to contacts with the outside and to control the staff and their late application in Italy have caused such a rapid spread of the disease in nursing homes which, affecting particularly fragile subjects, have led to unacceptable mortality rates. This negative experience led, after the approval of the anti-Covid-19 vaccines, to the production in Italy of some measures that specifically concerned the vaccination of subjects residing in nursing homes and the staff that work in who became part of the first categories subject to vaccination. This made it possible, during the second phase of the pandemic, to limit the contagion of these structures and significantly reduce mortality. Vaccines have a high social value because in addition to protecting the vaccinated person, they reduce the risk of contagion to the rest of the population. Vaccination is therefore a continuous balance between the individual and collective dimension of health in the spirit of mutual solidarity between the individual and the community. It is hoped that the proposals made in this article related to the Italian country can help systematize the processes to be implemented to protect the health of individuals as members of a close and fragile community.

## Data Availability

The study did not report any data of patients.

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
