# Peer review of "Nursing Home and Vaccination Consent: The Italian Perspective"

_vaccines, 2021, doi:10.3390/vaccines9050429_

Round 1
Reviewer 1 Report
Thank you very much for undertaking this important and timely research. The following is provided in the spirit of supporting the research.
Please state throughout whether the focus is on Italian law. Page 3 refers to Italian legal concepts. Please see the bottom of this review for a recommendation on how to frame this as a case study, based on Italy.
In the keywords section, you could add "capacity to consent", to capture a wider audience.
Under State of the Art on page 2, explain why these, but not other, countries were selected. Variations the response to covid vaccination for elders in aged care settings from select countries would also be informative.
Please clarify "prisoners" on page 2. Given the quote refers to elders, it seems odd that prisoners were included in the sentence that followed.
Pg 2 refers to Italian law. If the article is focused on Italian law, this should be stated in the title. And then compared with other select jurisdictions.
Clarify "grave forms" on pg 2.
Cites are needed for the claim that this strategy was adopted in "other nations of the European Union" and "most parts of the United States". Footnote 30 only refers to California, not the remaining states (or member states of the EU).
Please clarify "substantially unanimous".
Clarify "highest age group". Does this mean oldest? Largest? P 2.
On pg 3 please recognise that "advanced age" alone doesn't mean people are unable to express consent.
Throughout, please replace "capable of judgement" with "capacity to consent to vaccinations". Use of the concept of "capacity" will be better understood by your English-speaking audience. Use of the term will reduce ambiguity.
Also, please integrate literature on decisional capacity. A peson should be assessed for their ability to make a specific decision. Many countries have moved away from the global concept of capacity to the concept of decisional capacity, in keeping with human rights. ("Global concept of capacity" means that in some jurisdiction, a blanket assessment determines that the person is not capable of making a wide range of decisions. Increasingly, countries are focusing on decisional capacity-- a person's capacity to make a specific decision e.g. whether to ahve a vaccine). It would be important to express this development.
Current literature also refers to the practice of "supported decision-making". This replaces the historic concept of substituted decision-making and promotes the principles within the United Nations Convention on the Rights of Persons with Disabilities. Again, it is important to express this to your international audience.
Inclusion of the above two items requires additional research.
On p 3, cite the Western nations that require consent for people with limited or compromised capacity. (Or provide examples, with citations). The phrase "psychologial weakness" can be replaced with " limited or compromised capacity".
P3, the sentence begining with "Similarly..." can be rephrased for clarity.
P3, number 6: the phrase "remove their capacity" can be replaced with "remove their legal authority " to make their own health care decisions. (Capacity cannot be removed, as this sentence suggests).
The sentence that begins with "Case 2..." could be rephrased. Perhaps: Case 2 presents a dilemma, when a resident with capacity refuses the vaccination.
P4, replace "hosted in" with "residing in".
Rephrase the paragraph beginning with "Residents'..." for clarity.
Please provide a description, cite and date for the Oviedo Convention. Who are signatories to it?
I would recommend subheadings on page 3 and 4.
Where is the Alzheimer's Society located and is its document the foundation for the recommendations that follow? (Use "MCA" for the legislation the second time it is cited). P4.
There is an opportunity for critique. Is mental health legislation in one country (e.g. the MCA 2005), the optimum basis for vaccination consent in others? P4. Does the pandemic requires a customised legal approach?
For additional sources, please see Professor Ian Freckelton (Australia) and Professor Lawrence Gostin (USA). Both have published on public health law relating to covid.
Note: the goal is not to ascertain whehter a person in "possession of his or her full faculities" would have consented. Rather, the issue is whether a resident has met the capacity threshold to make the specific decision to have the vaccine.
The discussion of the conscious decision of a person to reside in a nursing home is problematic. The person may have capacity to make that decision, but not have the capacity to make a vaccination decision. And capacity can decline after admission to the nursing home. P 4.
Citations are needed for Italy's Decree. ideally, readers could access that on the internet.
Please reconsider the term word "assent". Its definition is legally distinct from the concept of "consent". P.4.
Restate "not always easy operative practice". Perhaps: "difficult to implement in practice". P4.
Option 2 appears to be coercive. It is important to ensure their human rights are protected (including social contact). See the Mandela Rules, the Principles for the Protection of Older Persons, and the United Nations Convention on the Rights of Persons with Disabilites. Also, recent WHO guidance. The article is silent on this crucial dimenstion. P 5.
Option 4: please clarify whether this relates to Italian law. And who would appeal on behalf of the person without capacity who has a guardian? What reforms are needed in these circusmtances to safeguard the resdient? e.g. independent advocacy to locate these residents and advocate for them to be vaccinated.
Option 6: replace the phrase "remove their capacity" because capacity cannot be removed. (See above re: this). P 5.
Clarify the sentence "Although different approaches...hermeneutics..." P 5.
Again, this section turns to Italian law. For an international audience, the Italian law has to be placed in context. The large paragraphs on P 5 can be subdivided into smaller paragraphs.
Rephrase "This is a case of..." and the remainder of the paragraph. Short, clear sentnces would aid the reader. P 5.
Please explain the "new right". When and why was it adopted? Is this beneifical for residents and therefore is there a lesson for the audience to be derived from this Italian "new right"? P 5.
(If you include workers in the analysis, please consider adding it to the title).
The ethical paragraph seemed to be an afterthought. NB: The competing concepts are autonomy and utilitarianism, I believe. P6. Reciprocity is a different concept. Reciprocity may be relevant if you argue that people can be contained in coercive settings, but there should in turn be a reciprocal health benefit for them. Again, articles by Professors Freckleton and Gostin may assist.
"Legal claims" can be replaced. P 6.
The conclusion is incomplete. It introduced new claims e.g. about the United Nations and lack of diligence. Abbreviations and "etc" need to be removed.
I suggest that this article be framed as a case study. And Italy is the case study. There are lessons to be learned from Italy and its legislation. Perhaps there are gaps, too, with regards to provision of vaccination for elders who decline vaccinations or do not have the capacity to consent. Perhaps those gaps are addressed by UN and WHO guidance on how to proceed when people lack capacity to consent to vaccinations. Pehaps other countries have developed benefical strategies. If the article is framed in this way, it would be clear why Italian law was cited and how it is relevant to a wider audience.
I wish you well with your article. And I wish you good health.
Please integrate
Author Response
1) Please state throughout whether the focus is on Italian law. Page 3 refers to Italian legal concepts. Please see the bottom of this review for a recommendation on how to frame this as a case study, based on Italy.
-Thanks, the missing information has been added.
2) In the keywords section, you could add "capacity to consent", to capture a wider audience.
-It has been added.
3) Under State of the Art on page 2, explain why these, but not other, countries were selected. Variations the response to covid vaccination for elders in aged care settings from select countries would also be informative.
- Thanks. At the beginning of the State of The Art we have added: “The following is the position of some European states, which together are representative of all the possible situations that exist in the remaining member states”.
- We haven’t data regarding “Variations the response to covid vaccination from select countries”.
4) Please clarify "prisoners" on page 2. Given the quote refers to elders, it seems odd that prisoners were included in the sentence that followed.
-Prisoners have been cited in the sentence by the National Healthcare System - Estrategia de vacunación frente a COVID19 en España, which states, as already indicated, “Residents and healthcare and social workers in nursing homes for the elderly and/or disabled. If necessary, priority goes to vaccinations for the most vulnerable residences (the greatest number of prisoners, poorest ability to adopt prevention and control measures and/or residences that have not had Covid-19 cases)”.
5) Pg 2 refers to Italian law. If the article is focused on Italian law, this should be stated in the title. And then compared with other select jurisdictions.
Clarify "grave forms" on pg 2.
-Good suggestion; we have corrected the title and changed “grave forms” with “severe forms of infectious disease”
6) Cites are needed for the claim that this strategy was adopted in "other nations of the European Union" and "most parts of the United States". Footnote 30 only refers to California, not the remaining states (or member states of the EU).
-We have changed the citation taking into consideration all the United States of America.
7) Please clarify "substantially unanimous".
- We have modified with “unanimous opinion”.
8) Clarify "highest age group". Does this mean oldest? Largest? P 2.
- It’s mean oldest. We have changed.
9) On pg 3 please recognise that "advanced age" alone doesn't mean people are unable to express consent.
-We have explained the concept better in this way: “In this context, obtaining informed consent to Covid-19 vaccination presents a considerable challenge, as the advanced age of nursing home residents and their frequent comorbidities may result in the inability to express their own consent”.
10) Throughout, please replace "capable of judgement" with "capacity to consent to vaccinations". Use of the concept of "capacity" will be better understood by your English-speaking audience. Use of the term will reduce ambiguity.
- Thank you for this advice. We have replaced the term with the one indicated.
11) Also, please integrate literature on decisional capacity. A peson should be assessed for their ability to make a specific decision. Many countries have moved away from the global concept of capacity to the concept of decisional capacity, in keeping with human rights. ("Global concept of capacity" means that in some jurisdiction, a blanket assessment determines that the person is not capable of making a wide range of decisions. Increasingly, countries are focusing on decisional capacity-- a person's capacity to make a specific decision e.g. whether to ahve a vaccine). It would be important to express this development.
- Thank you for the suggestion. We have supplemented the text.
12) Current literature also refers to the practice of "supported decision-making". This replaces the historic concept of substituted decision-making and promotes the principles within the United Nations Convention on the Rights of Persons with Disabilities. Again, it is important to express this to your international audience.
- Thank you for the food for thought. We have done.
13) Inclusion of the above two items requires additional research.
- We have added other research.
14) On p 3, cite the Western nations that require consent for people with limited or compromised capacity. (Or provide examples, with citations). The phrase "psychologial weakness" can be replaced with " limited or compromised capacity".
-Thank you for the suggestion. We have replaced with the one indicated.
15) P3, the sentence begining with "Similarly..." can be rephrased for clarity.
- Okay! We could write in this way “Similarly, assessing whether persons incapable of forming judgments are nevertheless capable of giving specific consent to vaccination may prove to be a complex challenge”.
16) P3, number 6: the phrase "remove their capacity" can be replaced with "remove their legal authority " to make their own health care decisions. (Capacity cannot be removed, as this sentence suggests).
- Thank you, we have replaced with “remove their legal authority”
17) The sentence that begins with "Case 2..." could be rephrased. Perhaps: Case 2 presents a dilemma, when a resident with capacity refuses the vaccination.
- Yes, we agree. We have modified it.
18) P4, replace "hosted in" with "residing in".
- Thank you. We have replaced it.
19) Rephrase the paragraph beginning with "Residents'..." for clarity.
- Thanks; we have rephresed in “residents who are weak-minded or have a fleeting memory due to initial cognitive impairment…”
20) Please provide a description, cite and date for the Oviedo Convention. Who are signatories to it?
- Thank you; we have cited Oviedo Convention.
21) I would recommend subheadings on page 3 and 4.
- Thank you for the tip. We have included subtitles.
22) Where is the Alzheimer's Society located and is its document the foundation for the recommendations that follow? (Use "MCA" for the legislation the second time it is cited). P4.
- We have provided an explanation in reference no. 53 “Alzheimer's Society is a United Kingdom care and research charity for people with dementia and their carers. It operates in England, Wales and Northern Ireland”.
23) There is an opportunity for critique. Is mental health legislation in one country (e.g. the MCA 2005), the optimum basis for vaccination consent in others? P4. Does the pandemic requires a customised legal approach?
- Thank you for your questions. Our thought is that what is provided in a country can be the basis for discussion, but each country has its own legislation and in these contexts you can not be "arbitrarily inspired" to the regulations of other countries. As for the "customised legal approach", there can be it as happened in Italy with the last decree on compulsory vaccinations for health personnel.
24) For additional sources, please see Professor Ian Freckelton (Australia) and Professor Lawrence Gostin (USA). Both have published on public health law relating to covid.
- Thanks for the suggestion. We have considered Gostin comment about “Mandating COVID-19 Vaccines” and added it in the references.
25) Note: the goal is not to ascertain whehter a person in "possession of his or her full faculities" would have consented. Rather, the issue is whether a resident has met the capacity threshold to make the specific decision to have the vaccine.
-Thanks, we recognized the indication.
26) The discussion of the conscious decision of a person to reside in a nursing home is problematic. The person may have capacity to make that decision, but not have the capacity to make a vaccination decision. And capacity can decline after admission to the nursing home. P 4.
- Thank you for the suggestion.
27) Citations are needed for Italy's Decree. ideally, readers could access that on the internet.
- Thank you for the clarification.
29) Restate "not always easy operative practice". Perhaps: "difficult to implement in practice". P4.
- Definitely! We have changed it.
30) Option 2 appears to be coercive. It is important to ensure their human rights are protected (including social contact). See the Mandela Rules, the Principles for the Protection of Older Persons, and the United Nations Convention on the Rights of Persons with Disabilites. Also, recent WHO guidance. The article is silent on this crucial dimenstion. P 5.
- Thank you for this advice. We have added on top of the page 4 this sentence: “Clearly, the residence where to move the "anti-vax" resident must ensure the protection of all rights of the person, in accordance with the provisions of Italian law and the "Convention on the Rights of Persons with Disabilities (CRPD)" of United Nations.”
31) Option 4: please clarify whether this relates to Italian law. And who would appeal on behalf of the person without capacity who has a guardian? What reforms are needed in these circusmtances to safeguard the resdient? e.g. independent advocacy to locate these residents and advocate for them to be vaccinated.
- We have modified several parts of the text, dividing it into further sub-paragraphs in which we have addressed the issues raised and added other pertinent reforms.
y
32) Option 6: replace the phrase "remove their capacity" because capacity cannot be removed. (See above re: this). P 5.
- Thank you. We have included “remove their legal authority”.
33) Clarify the sentence "Although different approaches...hermeneutics..." P 5.
- We have reformulated into” “The interpretations are varied and differentiated, however they are concentrated in two main orientations”.
34) Again, this section turns to Italian law. For an international audience, the Italian law has to be placed in context. The large paragraphs on P 5 can be subdivided into smaller paragraphs.
- Thank you. We have subdivided into other pragraphs.
35) Rephrase "This is a case of..." and the remainder of the paragraph. Short, clear sentences would aid the reader. P 5.
- Thanks, we have reformulated it; we have made also a punctuation change to make the concept clearer.
36) Please explain the "new right". When and why was it adopted? Is this beneifical for residents and therefore is there a lesson for the audience to be derived from this Italian "new right"? P 5.
(If you include workers in the analysis, please consider adding it to the title).
- Thank you. We have deleted the word “new”, “autonomy” and “reciprocity” that so as not to create interpretative misunderstandings.
37) The ethical paragraph seemed to be an afterthought. NB: The competing concepts are autonomy and utilitarianism, I believe. P6. Reciprocity is a different concept. Reciprocity may be relevant if you argue that people can be contained in coercive settings, but there should in turn be a reciprocal health benefit for them. Again, articles by Professors Freckleton and Gostin may assist.
- Thank you for the tip. Vaccination campaigns can not be based on the principle of utilitarianism, because the individual is not "sacrificed" for the benefit of the community, conversely, it is the community that is responsible for vaccination if the individual, due to health problems, can not be subjected to vaccination (herd immunity to preserve the immunologically incompetent).
As well explained in the opinion of the National Bioethics Committee, in the opinion called vaccines 1995, the obligation of vaccination is based not only on the right to health (art. 32 of the Constitution), but also on the duties of solidarity (art. 2 of the Italian Constitution).
Vaccines have a high social value because in addition to protecting the vaccinated person, they reduce the risk of contagion to the rest of the population. Vaccination is a continuous balance between the individual and collective dimension of health in the spirit of mutual solidarity between the individual and the community.
We have specified this concept in the manuscript; we have also added the term of “solidarity”.
38) "Legal claims" can be replaced. P 6.
- Thanks, we have reformulated the sentence.
39) The conclusion is incomplete. It introduced new claims e.g. about the United Nations and lack of diligence. Abbreviations and "etc" need to be removed.
- We have removed “etc” and the abbreviations. We have also modified the Conclusions.
40) I suggest that this article be framed as a case study. And Italy is the case study. There are lessons to be learned from Italy and its legislation. Perhaps there are gaps, too, with regards to provision of vaccination for elders who decline vaccinations or do not have the capacity to consent. Perhaps those gaps are addressed by UN and WHO guidance on how to proceed when people lack capacity to consent to vaccinations. Pehaps other countries have developed benefical strategies. If the article is framed in this way, it would be clear why Italian law was cited and how it is relevant to a wider audience.
- Thanks, the changes made take into account what is suggested
Reviewer 2 Report
The paper “Nursing home residents and vaccination consent” deals with a quite interesting and current issue. However, it needs some changes to make the paper itself suitable for publication.
First of all, the title which does not seem representative of the paper which is very focused on the Italian situation. Authors should change the title by making an explicit reference to this.
Paragraph 3, which is the central knot of the paper, sounds quite confusing and needs some clarifications.
- It is not clear if Authors refer to the Italian situation or if they are generally speaking about the issue of informed consent. It seems that the focus is on the italian landscape; please clarify it and make the point clearer.
- When dealing with the situation sub 3), Authors say that the “amministratore di sostegno” is specifically designated for health care issue. This affirmation may be a source of misunderstanding for readers. The Italian law 6/2004 which intriduces the “amministratore di sostegno” provides that the decree appointing the support administrator must contain an indication of the acts that the beneficiary can perform only with the assistance of the support administrator. As a consequence, this figure is not, always, designated for healthcare issues. Please, clarify.
- I dont’ agree with the use of the word “borderline” as used by Authors in the situation 5. Please, remove as it is also superfluous. These are subjects whose capacity to provide consent must be ascertained. Furthermore, I cannot see the difference with subjects sub 6). Both these subjects (sub 5 and 6) are not provided with a legal guardian and their capacity must be ascertained. Please, better clarify this point
- I’m really troubled with the part of this paragraph wher Authors propose the organization of “anti –vax” wards for residents. As it is set out by the Authors it sounds very like a kind of ghetto. Is there any literature evidence for this solution? Or it is a personal proposal by Authors themselves? I think that this point should be deepened and better motivated. Are similar solutions already be adopted? And with which results? Furthermore, at the beginning of this part, Authors affirm that COVID – 19 vaccination is not obligatory in the nations examined. Which nations?
- At the end of page 3, Authors introduce the Italian Legislative Decree January 5, 2021 that is, once again, recalled in the next page. This is confusing, so it must be modified. Furthermore, on the point the provisions of this decree are much more complex than what is stated at page 3. Since the Authors seem to have the aim of showing the Italian situation, this point should be deepened.
Paragraph 4.
This is a very interesting point, once again referred to the Italian situation. However, pending the revision of the article, an Italian Court (Court of Belluno) issued a very interesting judgement on this specific issue. Authors should comment on it.
Author Response
1) First of all, the title which does not seem representative of the paper which is very focused on the Italian situation. Authors should change the title by making an explicit reference to this.
- Thanks, we have done it.
2) Paragraph 3, which is the central knot of the paper, sounds quite confusing and needs some clarifications.
It is not clear if Authors refer to the Italian situation or if they are generally speaking about the issue of informed consent. It seems that the focus is on the italian landscape; please clarify it and make the point clearer.
- We have clearly explained the Italian situation, also citing what has been actually concretized in the international panorama, as well as dealing with the issue of informed consent in the light of the established and recent legislative interventions in Italy.
3) When dealing with the situation sub 3), Authors say that the “amministratore di sostegno” is specifically designated for health care issue. This affirmation may be a source of misunderstanding for readers. The Italian law 6/2004 which intriduces the “amministratore di sostegno” provides that the decree appointing the support administrator must contain an indication of the acts that the beneficiary can perform only with the assistance of the support administrator. As a consequence, this figure is not, always, designated for healthcare issues. Please, clarify.
- Thank you for the advice. We have changed the definition in brackes with “…if specifically designated for health issues in the decree of appointment of deeds…”.
4) I dont’ agree with the use of the word “borderline” as used by Authors in the situation 5. Please, remove as it is also superfluous. These are subjects whose capacity to provide consent must be ascertained. Furthermore, I cannot see the difference with subjects sub 6). Both these subjects (sub 5 and 6) are not provided with a legal guardian and their capacity must be ascertained. Please, better clarify this point
I’m really troubled with the part of this paragraph wher Authors propose the organization of “anti –vax” wards for residents. As it is set out by the Authors it sounds very like a kind of ghetto. Is there any literature evidence for this solution? Or it is a personal proposal by Authors themselves? I think that this point should be deepened and better motivated. Are similar solutions already be adopted? And with which results? Furthermore, at the beginning of this part, Authors affirm that COVID – 19 vaccination is not obligatory in the nations examined. Which nations?
-Thank you. We have deleted the word “borderline”. The nations examined are mentioned in the paragraph n.2
5) At the end of page 3, Authors introduce the Italian Legislative Decree January 5, 2021 that is, once again, recalled in the next page. This is confusing, so it must be modified. Furthermore, on the point the provisions of this decree are much more complex than what is stated at page 3. Since the Authors seem to have the aim of showing the Italian situation, this point should be deepened.
-Thanks, we have better explained the Italian situation adding the new decree of April 1, 2021 published in the Official Gazette.
6) Paragraph 4. This is a very interesting point, once again referred to the Italian situation. However, pending the revision of the article, an Italian Court (Court of Belluno) issued a very interesting judgement on this specific issue. Authors should comment on it.
-Thanks, we have done it.
Round 2
Reviewer 2 Report
I have carefully read the revised version of the paper and my concerns are quite unsolved. Many of my suggestions were not followed. Some examples. Paragraph 2 is still confusing since you have not, substantially, modified the paragraph, according to my suggestions. It is still not clear if you refer to the Italian situation or if you are generally speaking about the issue of informed consent. No substantial changes were made in the paragraph. I mean , whwn you list the situations challenging regarding the issue of informed consent, you speak about the "amministratore di sostegno", thus it seems that you refere to the Italian situation. This point is very confusing. Furthermore, sub point 5) of paragraph 3 the word "borderline" is still present. And you have not answered to my perplexity regarding the difference between subject under point 5) and those under point 6). Finally, regarding the item 5 of my report, you added some sentence on the Italian Decree 5 January , 2021 at the end of paragraph 3. It seems to me strange to speak about a decree when it was already mentioned twice in previous part of the paper. It adds to the confusion that reigns supreme in paragraph 3.
Author Response
In word
